# Tree Species Classification Based on ASDER and MALSTM-FCN

**Hongjian Luo** **, Dongping Ming \***, **Lu Xu and Xiao Ling**

School of Information Engineering, China University of Geosciences (Beijing), 29 Xueyuan Road,
Beijing 100083, China
*  Correspondence: mingdp@cugb.edu.cn

**Abstract:** Tree species classification based on multi-source remote sensing data is essential for ecological evaluation, environmental monitoring, and forest management. The optimization of classification features and the performance of classification methods are crucial to tree species classification. This paper proposes Angle-weighted Standard Deviation Elliptic Cross-merge Rate (ASDER) as a separability metric for feature optimization. ASDER uses mutual information to represent the separability metric and avoids the difficulty of differentiation caused by multiple ellipse centers and coordinate origins forming straight lines by angle weighting. In classification method, Multi-head Self-attention Long Short-Term Memory—Full Convolution Network (MALSTM-FCN) is constructed in this paper. MALSTM-FCN enhances the global correlation in time series and improves classification accuracy through a multi-head self-attention mechanism. This paper takes Beijing Olympic Forest Park (after this, referred to as Aosen) as the research area, constructs a tree species classification dataset based on an actual ground survey, and obtains a classification accuracy of 95.20% using the above method. This paper demonstrates the effectiveness of ASDER and MALSTM-FCN by comparing temporal entropy and LSTM-FCN and shows that the method has some practicality for tree species classification.

**Keywords:** tree species classification; ASDER; time-series classification; feature optimization; attention mechanism

## 1. Introduction

Remote sensing has the characteristics of full spatial coverage, multi-platform coordination, rapid change and update, and high-value density, which is an essential platform for Earth observation and provides decision support for land use, agricultural production, geological disasters, environmental protection, and other fields [1–4]. With the continuous optimization of sensor parameters, remote sensing began to develop in a "higher" direction, such as high spatial and spectral resolution. This presents both opportunities and challenges for land classification. High spatial resolution images can bring richer texture information, while over-dense textures can present difficulties in pixel classification [5,6]. Hyperspectral resolution images can provide rich spectral information, but selecting and utilizing spectral information is difficult for research. Improper image processing results in data redundancy, inefficient calculations, poor results, and so on [7]. With the increasing prominence of ecological problems, the management and maintenance of forest land are developing in the direction of refinement [8–10]. Species classification of forest land is a prerequisite for fine-grained management. Tree species classification is at the last level of the plant classification (i.e., family, genus, species) [11]. The finer the classification, the more difficult it is to distinguish. This paper will propose a convenient and fast chain of tree species classification techniques.

Tree species classification can be divided into non-time-series classification and time-series classification. Non-time-series classification is mostly conducted with commercial satellites such as high-resolution satellites and lidars. Although using data from these satellites may cause relatively high costs, it can finish the classification task excellently [12–14]. For

example, high-resolution images can better segment the patches, provide clear boundary information, and provide fine-grained boundary support for tree species classification [15,16]. In addition, hyperspectral satellites and lidars can provide rich feature information and enhance the difference between tree species [17–22]. Time-series classification is mostly based on medium-resolution satellites, with easy access to data and excellent classification results due to the full use of the tree species' phenological characteristics [23–25]. The multi-source data fusion approach can combine the advantages of different data to achieve better classification results [26–28]. It is feasible to classify tree species using boundary information of high-resolution images and time-series feature information of multispectral images. In addition, the key to multispectral image time-series classification lies in effective feature optimization and an efficient time-series classifier.

### 1.1. Feature Optimization and Separation Metrics in Tree Species Classification

Feature construction and feature optimization are tricky for tree species classification [29]. Common feature optimization methods include filtering, wrapping, and embedding methods. Tree species classification feature optimization mostly uses machine learning based wrapping methods and traditional filtering methods [15,17,24,26,30]. The filtering method is computationally simple and logically adequate. The filtering tree species feature optimization technique consists of a search algorithm and a separability metric [26].

The separability metric is the key to measuring the effectiveness of features and is one of the cores of the feature optimization algorithm. The variance and covariance are the most basic distinguishable measures, which can reflect the aggregation degree of the same class, and the differentiation degree of different classes. The variance and covariance are easy to calculate and can be easily integrated into feature optimization algorithms [31–33], and can also be directly used to build new feature optimization algorithms [34,35]. Distance-based separability metrics are an effective method to measure the effectiveness of features by calculating the magnitude of the distances of different samples in the feature space [36–38]. Distance-based separability metrics have a variety of types and can be flexibly selected. Among them, Euclidean distance measures the discriminative degree by calculating the linear distance in the feature space [39]; Hamming distance calculates the distinguishing degree by comparing the distance information on different dimensions [40]; Bhattacharyya distance and JM distance consider the overlapping degree of data distribution on the basis of distance calculation, which optimizes the interpretation of feature optimization and is more suitable for distinguishing difficult tasks [41,42]. The mutual information is the inverse expression of the differentiation degree and is a measure of the degree of data overlap. The smaller the mutual information, the higher the category differentiation. Mutual information is expressed as the degree of overlap in areas in two-dimensional space and in higher-dimensional space as the overlap in the corresponding space [43]. By optimizing the calculation of mutual information in different feature spaces, the effect of feature optimization can be improved [43,44].

The advantages of the above separability metrics are interpretability and generalizability, which can be used as stable metrics for different feature spaces corresponding to different classification tasks and can be well generalized to the computation of higher dimensional spaces. The disadvantage lies in the lack of targeted metrics for specific feature construction algorithms.

### 1.2. Tree Species Time-Series Classifier and Deep Learning Time-Series Model

Machine learning methods are mostly adopted in the current time-series tree species classification [24,45,46]. Among them, the RF algorithm has been the most widely used and has better applicability compared to methods such as Support Vector Machines (SVM) [23,25,47,48]. The advantages of machine learning methods are low computational effort, robust classification results, and high interpretability. The disadvantages are the difficulty in obtaining deep features, poor scalability of the classifier, and difficulty in improving the classification accuracy. Deep neural networks such as LSTM and Convolutional



Neural Network (CNN) can achieve higher classification accuracy [49,50]. The development of high-performance GPUs has reduced the computational time of deep learning, while the exploration of deep learning in computer vision and natural language processing has enhanced the interpretability of deep learning [51,52]. This laid the theoretical foundation and hardware foundation for the time-series classification of tree species based on deep learning.

Proper architecture is the key to deep learning to deal with time-series problems. Multilayer Perceptron (MLP) is one of the classical neural network models, which processes data through a series of fully connected layers [53–55]. The problem with using MLP to deal with time-series is that it can only deal with univariate time-series or flattened multivariate time-series and cannot obtain the deep semantics between multivariate variables. The advantage of MLP is that all the information can be fed into the network at once, which can better focus on global information. LSTM is a variant of Recurrent Neural Network (RNN), which solves the problem of long-term sequence information forgetting by cleverly designing memory units [56]. The directivity of the memory unit means LSTM can only learn in one direction, and it is easy to ignore global information. CNN is computed through a sliding window, and 1D-CNN has good results in time-series problems [57]. Enhanced by multi-channel and multi-scale transformations, CNN has become one of the essential tools in time-series classification [58,59]. The attention mechanism is a neural network that abandons the above architecture and only considers the correlation among data, which can input global information at once and calculate the global correlation [60]. A satisfactory network effect is often an effective combination of several network architectures [61,62].

In summary, there are two research focuses on the time-series classification of tree species based on multispectral images:

1. Creating targeted separability metrics for tree species classification feature optimization based on the desired feature construction method;
2. Using deep learning, which has the advantages of strong scalability and the ability to mine deep information, to construct a reasonable time series classifier and improve the accuracy of tree species classification.

Optimizing time-series features helps to refine the classification [63], and a reasonable deep learning method can improve the classification accuracy [64–66]. This study improves the classification accuracy of tree species based on feature optimization and deep learning methods. In this paper, we propose ASDER, a separability metric for the normalized difference method, to determine the separability of tree species features. In addition, this paper modifies part of the structure based on the existing network model and proposes the MALSTM-FCN model, which can better adapt to tree species classification tasks with different features.

## 2. Research Area and Data

### 2.1. Research Area and Sample Collection

#### 2.1.1. Research Area Introduction

Beijing is located at the northwestern edge of the North China Plain. It has a typical warm temperate, semi-humid, continental monsoon climate with four distinct seasons, high temperatures, and rain in summer. Beijing Olympic Forest Park is located at the northern end of the central axis of Beijing city, starting from the central Olympic area in the south and bordering the suburban protective green area of Qinghe North Road in the north, straddling the main road of the North Fifth Ring Road, which is the main urban park for leisure and sports of urban residents in the northern region of Beijing, as shown in Figure 1. The park covers a total area of 680 hm$^2$, including a green area of about 478 hm$^2$ and a water surface area of 67.7 hm$^2$, with a green coverage rate of 95.61%. Aosen is rich in forest resources and dominated by trees and shrubs, making it an ideal area for this research.

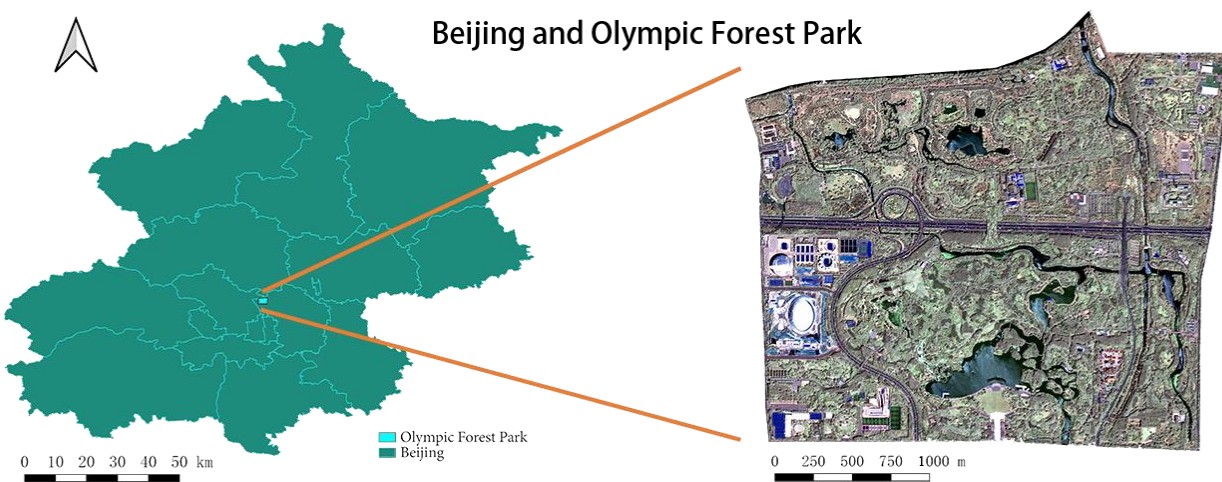

**Figure 1.** Research area.

As of 2008, there were over 100 species of trees totaling approximately 530,000, 80 species of shrubs, and 100 species of ground cover plants in Aosen, forming a natural forest system classified according to biodiversity. The trees include Chinese pine (*Pinus tabuliformis* Carriere), juniper (*Juniperus chinensis* L.), sophora (*Styphnolobium japonicum* (L.) Schott), Chinese willow (*Salix matsudana* Koidz.), toon (*Ailanthus altissima* (Mill.) Swingle), and koelreuteria (*Koelreuteria paniculata* Laxm.) [67].

### 2.1.2. Sample Collection

The quality of the dataset will directly affect the accuracy of the classification results [68,69]. Due to the limited visual discrimination of tree species on meter-scale true color images, tree species cannot be determined directly from images. In this paper, ground truthing is carried out based on positioning techniques. Some photos of the sampling process and the sharpened image photos are shown in Figure 2.

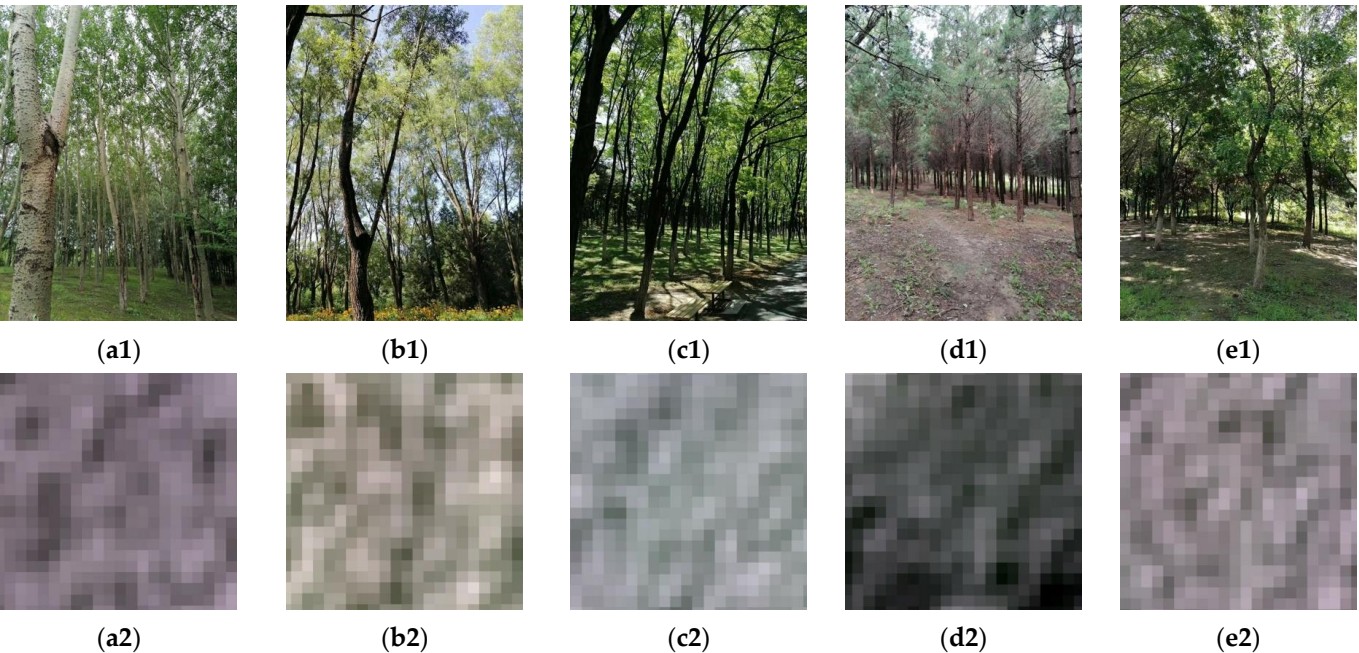

**Figure 2.** Field sampling photos and high-resolution fusion image photos. Where **a1** and **a2** are populus, **b1** and **b2** are willow, **c1** and **c2** are sophora, **d1** and **d2** are pine, **e1** and **e2** are ash. They are the main tree species in each class.

Through a field survey, a total of 2104 sample points were recorded in this paper. As shown in Figure 3, they were classified into six classes to obtain Populus samples, Willow samples, Sophora samples, Pine samples, Other tree samples, and Grassland samples (negative samples). The principle of classification is the merging of like items. For example, Weeping willow and Chinese willow are combined into the Willow Class, and Masson's pine and Chinese red pine are combined into the Pine Class. In addition, those with smaller sample numbers were uniformly grouped into one class, such as Chinese toon and Chinese ash. The specific number of each classification is shown in Table 1.

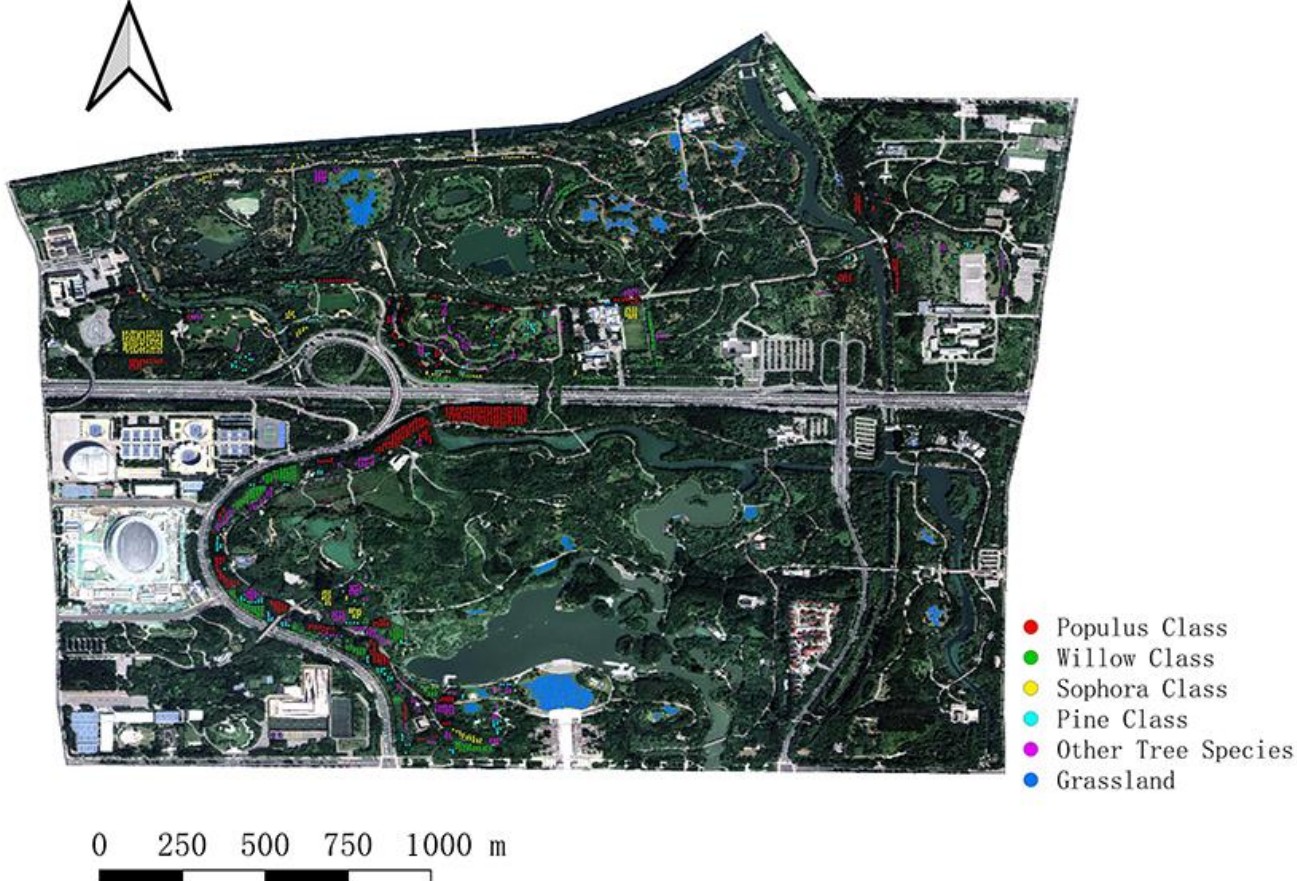

**Figure 3.** Sample point locations.

## 2.2. Introduction of Remote Sensing Data

Gaofen 2 (GF2) satellite is the first civil optical remote sensing satellite with better than 1-m spatial resolution developed independently by China, equipped with two high-resolution 1-m panchromatic and 4-m multispectral cameras with sub-meter spatial resolution, high positioning accuracy, and fast attitude maneuverability, effectively enhancing the comprehensive observation effectiveness of the satellite. Some areas of Aosen show fragmentation of tree patches and fine-grained characteristics. This poses difficulties for object-based classification. The results of 1-m panchromatic fused images using GF2 can effectively extract tree species patches and support the classification work. Due to the problem of temporal phase between tree species, such as the similar spectral properties of winter deciduous vegetation that are difficult to distinguish, the GF2 image of 28 May 2021, was selected for the patch calculation in this paper.

**Table 1.** The number of sample points in each class.

| Classes | Include | Number |
|---|---|---|
| Populus Class | Chinese white poplar (*Populus tomentosa*) *<br>Korean white poplar (*Populus davidiana* Dode)<br>White poplar (*Populus alba* L.) | 514 |
| Willow Class | Weeping willow (*Salix babylonica* L.)<br>Chinese willow (*Salix matsudana* Koidz) | 300 |
| Sophora Class | Japanese pagoda tree (*Sophora japonica* Linn. var.<br>japonica f. oligophylla Franch.)<br>Jinye pagoda tree (*Sophora japonica* cv. jinye)<br>Black locust (*Robinia pseudoacacia* L.) | 249 |
| Pine Class | Masson's pine (*Pinus massoniana* Lamb.)<br>Chinese red pine (*Pinus tabuliformis* Carr.) | 178 |
| Other Tree Species | Chinese ash (*Fraxinus chinensis* Roxb.)<br>Tree of heaven (*Ailanthus altissima*)<br>Chinese toon (*Toona sinensis* (A. Juss.) Roem.)<br>Else | 384 |
| Grassland | | 479 |

* To avoid ambiguity, the Latin scientific name is given in parentheses after each tree species.

The Sentinel 2 (S2) satellite is an Earth observation mission under the Copernicus program of the European Space Agency, which focuses on observations of the Earth's surface to provide relevant telemetry services, such as forest monitoring, land cover change detection, and natural hazard management [70]. The program is a constellation of two identical satellites (A and B). S2 carries a multispectral imager that can cover 13 spectral bands with an amplitude of 290 km. The data products of S2 range from visible to short-wave infrared with different spatial resolutions, among which the spatial resolution of RGB and NIR bands is 10 m, which basically meets the experimental needs. For tree species classification, S2 data products contain three red-edge bands before the NIR band (835.1 nm (S2A)/833 nm (S2B)) and one red-edge band after the NIR band, which can effectively enhance the spectral differentiation [23]. Its red-edge band position is shown in Figure 4. In this paper, the S2 product data were obtained by Google Earth Engine (GEE) [71]. A total of 48 S2 product images were acquired (Table 2), and 10 bands were selected for feature calculation in each image, namely Blue, Green, Red, Red Edge1, Red Edge2, Red Edge3, NIR, Red Edge4, SWIR1, and SWIR2.

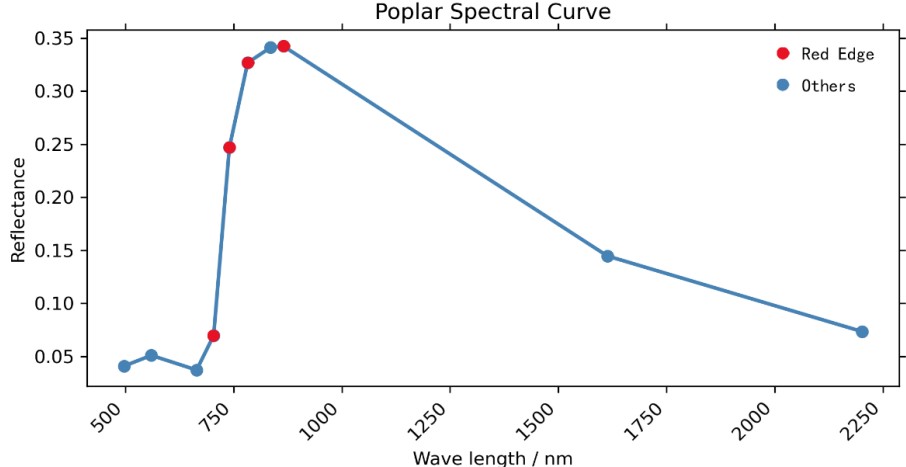

**Figure 4.** Location of the red edges in the S2 spectrum (with the A-star Populus Class as an example).

**Table 2.** Table of S2 image time-series.

| Date of Data | | | | | | | |
|---|---|---|---|---|---|---|---|
| 2021/1/2 | 2021/1/24 | 2021/3/20 | 2021/5/2 | 2021/8/2 | 2021/10/1 | 2021/10/26 | 2021/12/3 |
| 2021/1/4 | 2021/1/29 | 2021/3/25 | 2021/5/7 | 2021/8/10 | 2021/10/11 | 2021/11/10 | 2021/12/5 |
| 2021/1/9 | 2021/2/3 | 2021/4/7 | 2021/5/29 | 2021/8/17 | 2021/10/16 | 2021/11/13 | 2021/12/13 |
| 2021/1/12 | 2021/2/6 | 2021/4/14 | 2021/6/1 | 2021/8/27 | 2021/10/19 | 2021/11/23 | 2021/12/18 |
| 2021/1/17 | 2021/2/8 | 2021/4/17 | 2021/6/3 | 2021/9/1 | 2021/10/21 | 2021/11/25 | 2021/12/20 |
| 2021/1/22 | 2021/2/26 | 2021/4/19 | 2021/6/21 | 2021/9/29 | 2021/10/24 | 2021/11/30 | 2021/12/30 |

## 3. Methods

In this paper, tree species time-series features are extracted by sample points, and these time-series features are calculated from different combinations of bands by the normalized difference method. In addition, feature optimization is performed using an Exhaustive Feature Search (EFS) method based on ASDER. Then, the screened time-series features are input into MALSTM-FCN for training. Finally, the model is used to classify the tree species. Figure 5 shows the technical flow diagram of this Methods section, which contains three main parts. The first part is image preprocessing and segmentation, which mainly includes image correction, fusion, image segmentation, and generation of centroids. The second part is the feature optimization based on EFS + ASDER, which constructs features by exhausting all band combinations and sorts all features under all time phases by ASDER to filter out the feature indices required for classification. The third part is about constructing the MALSTM-FCN model, and the training and prediction using the constructed model. In this paper, the second and third parts will be explicitly described.

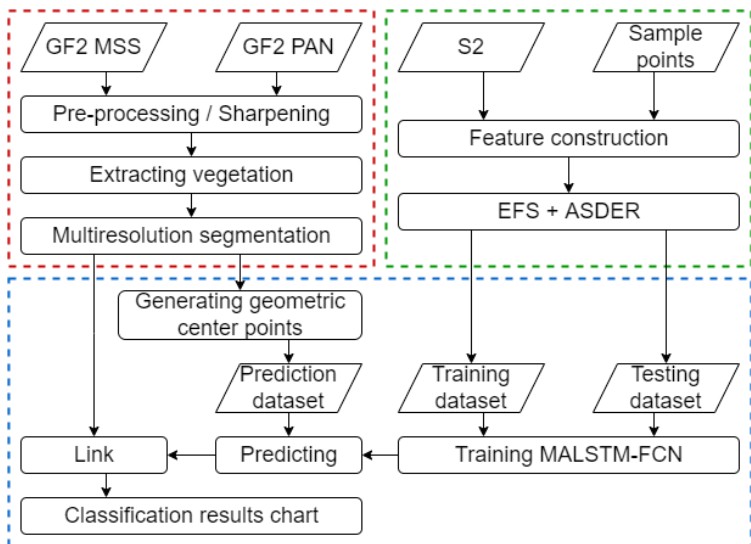

**Figure 5.** Technical flow diagram. The first part is in the red box, the second is in the green box, and the third is in the blue box.

### 3.1. Image Pre-Processing and Image Segmentation

In this paper, radiometric calibration, atmospheric correction, and orthorectification are applied to the PAN and MSS of GF2, and the image is sharpened using the Gram-Schmidt (GS) method. The advantage of GS is that the spatial texture information is better preserved, which is very important for the following segmentation work. Then, this paper performs binary classification of the fused images by SVM to extract vegetation as foreground information. The distribution pattern of tree species within the Aosen is fragmented, diverse, and multi-scale, as obtained through Google high-definition imagery. We use the Multiresolution Segmentation (MRS) method in eCognition software to segment

the foreground information. MRS can improve the efficiency and accuracy of segmentation while retaining detailed information.

The spatial resolution of different bands in the S2 product used in this paper is not consistent. In this paper, all the bands involved in feature construction are resampled with a resolution of 10 m to guarantee that the constructed features are completely uniform in spatial location and resolution.

### 3.2. ASDER-Based Feature Optimization

### 3.2.1. Normalized Difference Method and Separability Metrics

The normalized difference method is a commonly used feature construction method with the advantages of easy calculation and obvious distinction [72]. For deep neural networks, the normalized difference method can avoid the preliminary data normalization operation and facilitate the research, which is calculated as follows:

$$\text{ND} = \frac{\text{Band\_first} - \text{Band\_second}}{\text{Band\_first} + \text{Band\_second}}, \tag{1}$$

At this point, the problem of feature separation can be transformed into a coordinate calculation problem in the Cartesian coordinate system where Band_first and Band_second are used as *X* and *Y* axis coordinates, respectively.

Traditional separability metrics include Marxian distance, JM distance, scatter, variance, and covariance [73–80]. Most of such methods calculate the discriminant based on the distances of different classes in the feature space, which has a certain universality and lacks relevance. The reasons are as follows:

1. The variance can only evaluate the separation in one dimension and cannot take into account the overall spatial information;
2. The distance-based separation calculation does not take direction into account very well.

As a result, ASDER is proposed in this paper as a separability metric for the normalized difference method.

### 3.2.2. Introduction to ASDER

The essence of the ASDER method is a mutual information calculation method based on a standard deviation ellipse. The standard deviation ellipse constructs and draws an ellipse by calculating parameters such as variance and covariance between spatial points. The range contained in the ellipse is the region with the densest distribution of spatial points [81–84]. By calculating the percentage of the overlap area between the standard deviation ellipses constructed by different classes of sample points, the separation index ASDER is obtained under this band combination, which expresses the effectiveness of the band combination qualitatively and quantitatively. The formula for calculating ASDER is as follows:

$$\text{ASDER} = \frac{2 \cdot \sum\limits_{i=1}^{m-1} \sum\limits_{j=i+1}^{m} E_{i \cap j}}{\widetilde{A} \cdot (m-1) \cdot \sum\limits_{i=1}^{m} E_i}, \tag{2}$$

where *m* is the number of classes in the classification, $\widetilde{A}$ is the angular mean, *E* is the area of the standard deviation ellipse, and $E_{i \cap j}$ represents the intersection area of two standard deviation ellipses, as shown in Figure 6a. The following equation can calculate the standard deviation ellipse:

$$\sigma_x = \sqrt{\sigma} \sqrt{\frac{\sum\limits_{i=1}^{n} (\widetilde{x}_i \cos\theta - \widetilde{y}_i \sin\theta)^2}{n}},$$
$$\sigma_y = \sqrt{\sigma} \sqrt{\frac{\sum\limits_{i=1}^{n} (\widetilde{x}_i \sin\theta + \widetilde{y}_i \cos\theta)^2}{n}}, \tag{3}$$

$$\theta = -\frac{\arctan\left(\frac{A+B}{C}\right)}{\pi} \cdot 180, \; A = n \cdot (c_{xx} - c_{yy}), \; C = 2n \cdot c_{xy}, \; B = \sqrt{A^2 + C^2}, \quad (4)$$

$$\mathrm{cov}(x,y) = \begin{pmatrix} c_{xx} & c_{xy} \\ c_{yx} & c_{yy} \end{pmatrix} \quad (5)$$

where $\sigma_x$ and $\sigma_y$ denote the long and short axes of the ellipse. $\theta$ is the rotation angle, positive counterclockwise, starting from the positive half-axis of the $x$ axis in the two-dimensional Cartesian coordinate system. $n$ is the number of sample points. $\tilde{x}$ and $\tilde{y}$ denote the difference between the $x$ and $y$ coordinates and their respective arithmetic means. $\sigma$ denotes the standard deviation level, which is three levels with values of 2, 8, and 18. $c_{xx}$, $c_{yy}$ and $c_{xy}$ are obtained from the covariance.

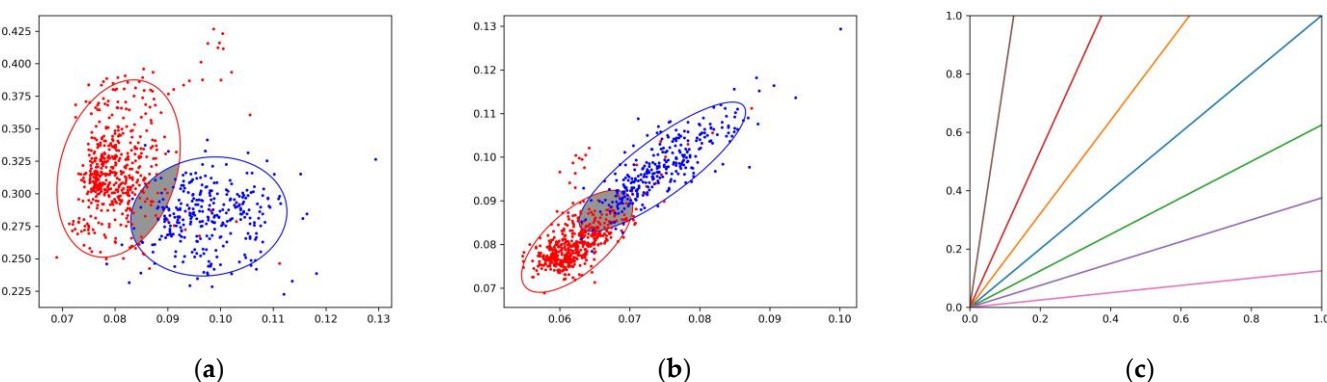

(a)   (b)   (c)

**Figure 6.** Schematic chart of a standard deviation ellipse. The red and blue dots and circles in (**a**) and (**b**) represent the two types of sample points and the standard deviation ellipse they form, and the gray part indicates the overlapping area. (**a**) is the normal intersection of 2 standard deviation ellipses, (**b**) appears a particular case where the centers of the two standard deviation ellipses and the origin almost course a straight line, and (**c**) are some straight lines through the origin, where the indices constructed by the normalized interpolation method have the same value on the same line.

This paper obtains the standard deviation ellipse by plotting the arithmetic average of all coordinate points as the circle's center. However, if the intersection of standard deviation ellipses is used solely as the basis for judgment, a particular case is likely to occur (Figure 6b). In this case, the two standard deviation ellipses appear on the same line as the coordinates' origin. It is learned by the normalized difference method that the index values calculated on the straight line over the coordinate source are the same (Figure 6c). If the scatter points of different classes can almost form a straight line with the origin, it is impossible to distinguish the difference between them, thus affecting the actual classification effect.

To avoid the abovementioned problem, this paper introduces the angular mean as the weight and uses the standard deviation of level three for the calculation to prevent the non-overlap phenomenon. $\widetilde{A}$ is calculated as follows:

$$\widetilde{A} = \frac{\sum\limits_{i=1}^{m-1} \sum\limits_{j=i+1}^{m} A_{ij}}{\sum\limits_{i=1}^{m-1} i}, \quad (6)$$

where $A_{ij}$ denotes the angle formed with the center and coordinate origin of the two standard deviation ellipses; by introducing angle weights, the ASDER is weighted to infinity if there is a linear relationship with different circle centers and origins.

### 3.3. MALSTM-FCN Tree Species Time-Series Classification

The LSTM-FCN model was proposed by Karim in 2018 and has been widely used in time-series classification [85–89]. FCN is a powerful model for dealing with the time-series problem [61]. It is based on a three-layer 1D-CNN model and achieves good classification results, whereas LSTM, as another branch, suffers from the problem of insufficient global consideration.

Transformer, and its subsequent variants, mainly benefits from the encoder and decoder architecture in Transformer [51,52,62]. The multi-head self-attention mechanism is the core of the encoder and decoder. It provides a method to calculate the correlation between all variables at once, which can effectively compensate for the shortage of LSTM. Based on this, this paper proposes the network architecture of MALSTM-FCN by adding the multi-head self-attention mechanism behind the LSTM branch to improve globalization. Figure 7 shows the network architecture.

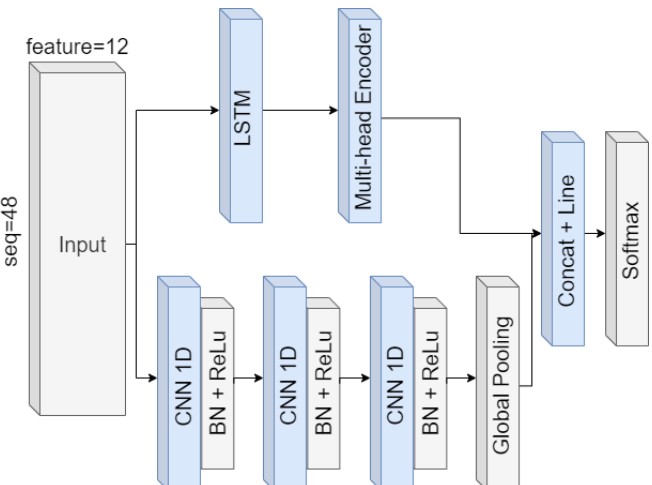

**Figure 7.** MALSTM-FCN structure diagram.

In this paper, the improvement process is based on the assumption that there is an interconnection between features of different time phases. So after LSTM branching, attention encoding is performed. Since the time-series classification is a sequence-to-class (not sequence-to-sequence) correspondence, the calculation is performed using the self-attention approach. The schematic diagram of multi-head self-attention is shown in Figure 8.

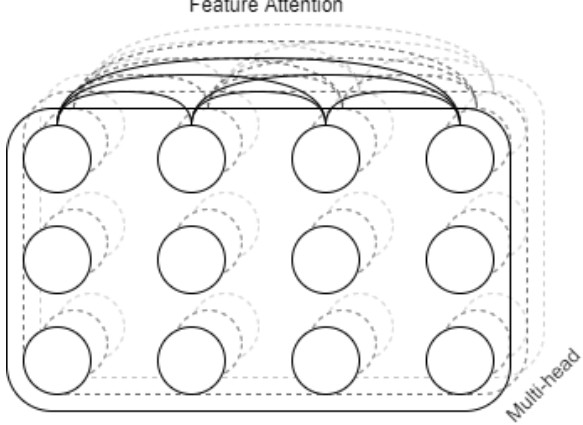

**Figure 8.** Diagram of Multi-head.

The self-attention mechanism between features under different time phases is calculated, and the output of the last time phase is used as the result. At this time, the output of the final temporal phase can focus on all features of all temporal phases by the attention mechanism, thus enhancing global consideration and strengthening generalization ability and compensate for the shortage of LSTM.

*3.4. Accuracy Assessment*

The confusion matrix is an effective way to measure the accuracy of a classification task. In the confusion matrix, each row represents the true classification, and each column represents the predicted classification. The values on the diagonal of the matrix indicate the number of correctly classified samples, while the values on the non-diagonal indicate the number of misclassified samples. The overall classification accuracy is calculated as follows:

$$\text{Acc} = \frac{\text{TP}}{\text{TP} + \text{FP}}, \tag{7}$$

where TP represents the number of correct classifications and FP represents the number of incorrect classifications.

## 4. Experiment

*4.1. Spectral Feature Optimization*

In this experiment, tree species were divided into five classes, of which the first four classes were distinguished individually; namely, Populus, Willow, Sophora, and Pine. In this paper, separation experiments were conducted for the four classes of tree species. Table 3 was obtained by calculating the ASDER indices of different band combinations under different time phases.

**Table 3.** Table of ASDER values for various combinations.

| Combination of Tree Species | Time Phase | Band First | Band Second | Rate (%) | Angle (°) |
|---|---|---|---|---|---|
| Populus Class AND Willow Class | 2021/5/2 | Band_3 | Band_8 | **1.235** | 5.225 |
| | 2021/5/2 | Band_3 | Band_7 | 1.326 | **5.476** |
| | 2021/4/17 | Band_3 | Band_8 | 1.389 | 4.914 |
| Populus Class AND Sophora Class | 2021/4/17 | Band_3 | Band_8 | **0.000** | **9.385** |
| | 2021/4/19 | Band_3 | Band_8 | 0.000 | 9.385 |
| | 2021/4/17 | Band_3 | Band_7 | 0.041 | 9.294 |
| Populus Class AND Pine Class | 2021/12/20 | Band_2 | Band_8 | **0.784** | 8.481 |
| | 2021/12/13 | Band_2 | Band_8 | 0.792 | 8.704 |
| | 2021/12/13 | Band_2 | Band_6 | 0.864 | **11.524** |
| Willow Class AND Sophora Class | 2021/4/7 | Band_8 | Band_12 | **1.201** | **12.303** |
| | 2021/4/7 | Band_6 | Band_12 | 1.334 | 11.866 |
| | 2021/4/14 | Band_8 | Band_12 | 1.420 | 12.086 |
| Willow Class AND Pine Class | 2021/2/3 | Band_4 | Band_6 | **1.695** | **12.514** |
| | 2021/1/29 | Band_4 | Band_6 | 1.944 | 11.270 |
| | 2021/2/6 | Band_4 | Band_6 | 1.965 | 10.680 |
| Sophora Class AND Pine Class | 2021/12/30 | Band_4 | Band_8 | **0.563** | 12.674 |
| | 2021/2/6 | Band_4 | Band_8 | 0.640 | 11.445 |
| | 2021/2/3 | Band_4 | Band_6 | 0.696 | **14.573** |

The bolded part represents the best value under the current experiment.

Where Band_2–Band_8, Band_11, and Band_12 represent the corresponding bands in S2, respectively. By calculating the ASDER of different tree species combinations under different time phases and band combinations, this paper lists the top three ASDER results. It is clear from the analysis that for the classification of poplar and willow species and poplar and sophora species, the green band is important. A good differentiation effect is

achieved by combining the green band and the NIR band. For the classification of poplar and pine, the combination of the blue and NIR bands can improve the differentiation. The second short-wave infrared band was significant for the classification of willow and sophora. Water has a strong spectral absorption peak in the short-wave infrared band, and combining it with the near-infrared or red-edge band makes the differentiation of tree species higher. For willow and pine species, using the red-edge band instead of the NIR band gives the desired differentiation index. In addition, the best results for sophora and pine species were obtained with the conventional NDVI.

In this paper, the validity of ASDER is verified by comparing the misclassification results of NDVI and ASDER top-ranking indices between corresponding tree species using NDVI as the benchmark. This paper uses the classical LSTM network as a classifier and is trained for 500 epochs. The comparison results obtained are shown in Table 4.

**Table 4.** Comparison of band combinations.

| First Species | Second Species | Band First | Band Second | FP Rate (%) | FN Rate (%) |
|---|---|---|---|---|---|
| Populus Class | Willow Class | Band_3 | Band_8 | 5.28 | 6.19 |
| | | Band_4 | Band_8 | 8.06 | 16.19 |
| Populus Class | Sophora Class | Band_3 | Band_8 | 1.39 | 1.15 |
| | | Band_4 | Band_8 | 0.83 | 0.00 |
| Populus Class | Pine Class | Band_2 | Band_8 | 0.30 | 4.01 |
| | | Band_4 | Band_8 | 0.30 | 3.21 |
| Willow Class | Sophora Class | Band_8 | Band_12 | 10.95 | 3.44 |
| | | Band_4 | Band_8 | 12.86 | 4.01 |
| Willow Class | Pine Class | Band_4 | Band_6 | 7.62 | 24.08 |
| | | Band_4 | Band_8 | 10.95 | 24.88 |

FP is the number of Second species misclassified into First species, while FN is the opposite. The ratio of the total number of misclassifications was obtained by multiple experiments and summed up in this paper. In the classification comparisons between poplar and willow, willow and sophora, and willow and pine, ASDER's best index yielded better classification results (Table 4). This also proves the rationality of ASDER.

Finally, this paper screened the top three band combinations that distinguish between different tree species, a total of 18 (Table 5). A total of 12 different wave combinations are obtained after eliminating redundancy, and 12 feature indices are constructed by the normalized difference method. Finally, the 12 time-series feature indices of the final input network are obtained by time-series extraction.

**Table 5.** Band combinations for the constructed normalized difference indices.

| Combination of Tree Species | Band First | Band Second | Combination of Tree Species | Band First | Band Second |
|---|---|---|---|---|---|
| Populus Class AND Willow Class | Band_3 | Band_8 | Willow Class AND Sophora Class | Band_8 | Band_12 |
| | Band_3 | Band_7 | | Band_6 | Band_12 |
| | Band_3 | Band_6 | | Band_4 | Band_8 |
| Populus Class AND Sophora Class | Band_3 | Band_8 | Willow Class AND Pine Class | Band_4 | Band_6 |
| | Band_3 | Band_7 | | Band_4 | Band_5 |
| | Band_8 | Band_11 | | Band_3 | Band_6 |
| Populus Class AND Pine Class | Band_4 | Band_6 | Sophora Class AND Pine Class | Band_4 | Band_8 |
| | Band_8 | Band_11 | | Band_4 | Band_6 |
| | Band_3 | Band_6 | | Band_4 | Band_7 |

### 4.2. MALSTM-FCN Tree Species Classification Results

The experimental environment of MALSTM-FCN is the Pytorch framework based on NVIDIA GPU. The initial learning rate is $1 \times 10^{-3}$ for 300 epochs of training, and the learning rate becomes $1 \times 10^{-4}$ after 200. The dataset introduced in Section 3.1 is randomly split in the ratio of 7:3 in this paper, with 70% as the training dataset and 30% as the testing dataset.

The overall accuracy of the testing dataset judges the results of the training. The average accuracy of MALSTM-FCN is above 95%. Figure 9 shows the final classification results using the trained model. The image map on the right shows the classification results, the vegetation segmentation by GF2, and the high-definition image map by Google. The specific effect of tree species classification cannot be judged by visual interpretation alone. This paper observes the specific classification effect by calculating the confusion matrix to understand the classification results more precisely.

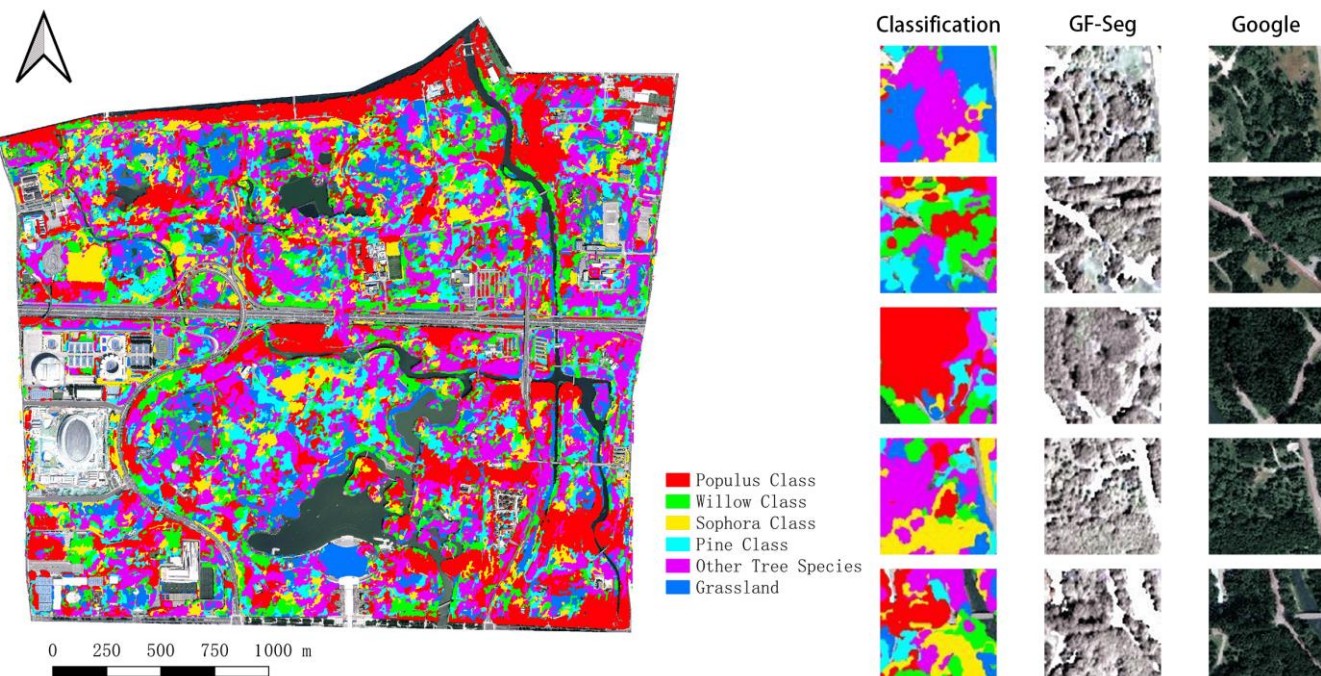

**Figure 9.** Classification results.

The following results are obtained in this paper by analyzing the calculated values of the confusion matrix (Figure 10). Grasslands as the background was perfectly extracted, and the classification accuracy of each tree species itself was over 90%. The misclassified parts were concentrated in the Pine Class (Pi) and Other Tree Species (O). The most misclassifications were found between Sophora Class (S) and O. The reasons for misclassification are as follows. Firstly, O contains multiple tree species with complex spectral structures, which are easy to cross with other classes, leading to misclassification. Secondly, the S2 pixel itself has the phenomenon of confusion, which easily leads to misclassification. In general, the tree species time-series classification based on MALSTM-FCN has achieved good results.

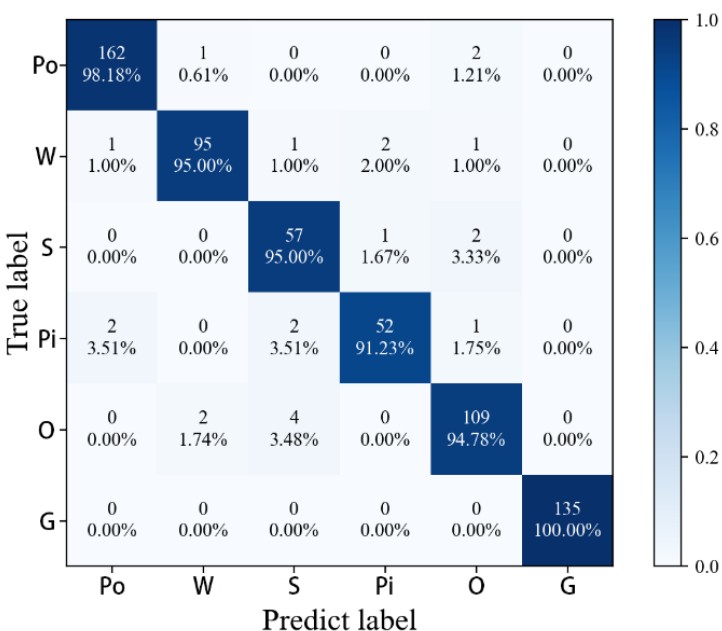

**Figure 10.** MALSTM-FCN confusion matrix.

## 5. Discussion

### 5.1. Validity of ASDER

This paper further explores the constructed indices by calculating other characteristics of the 12 band combinations. In this paper, the separation of tree species means at different time phases was calculated by sliding windows. The calculation results are shown in Table 6 where var max is the value with the most significant variance in a year and represents the moment when the separation ability of tree species is the largest. Var sum and entropy sum represent the sum of variance and the sum of entropy. The analysis in the table shows that the maximum separation under a single temporal phase occurs in the band combination of 2–6, while the overall best combination is 4–7, and the single and the overall lowest combination is 4–5. The results show that the band combination of 4–5 has insufficient separation on general temporal phases. In addition, it performs well on ASDER, especially for willow and pine samples with high separation (Table 5).

**Table 6.** Table of variance, entropy values.

| Band Combine | Var Max | Var Sum | Entropy Sum | Band Combine | Var Max | Var Sum | Entropy Sum |
|---|---|---|---|---|---|---|---|
| Band_2–6 | **0.043** * | 0.305 | 3.870 | Band_4–6 | 0.028 | 0.392 | 4.501 |
| Band_2–8 | 0.038 | 0.261 | 3.535 | Band_4–7 | 0.029 | **0.398** | **4.593** |
| Band_3–6 | 0.029 | 0.263 | 3.150 | Band_4–8 | 0.028 | 0.345 | 4.312 |
| Band_3–7 | 0.030 | 0.277 | 3.333 | Band_6–12 | 0.020 | 0.384 | 3.787 |
| Band_3–8 | 0.027 | 0.238 | 2.993 | Band_8–11 | 0.019 | 0.316 | 2.832 |
| Band_4–5 | **0.017** | **0.181** | **2.552** | Band_8–12 | 0.018 | 0.378 | 3.538 |

* The bolded part represents the maximum and minimum values under the current experiment.

Other characteristics of the 4–5 band combination and their rankings are further calculated in this paper. Using 2021-04-14 as the time phase, the distance (Euclidean distance) sum of the tree species mean points is 0.5465, ranking 39th, the variance on the *x*-axis is 22nd, the variance on the *y*-axis is 42nd, and the covariance is 19th. It can be concluded that the 4–5 band combination is not effective under the traditional discrimination measure.

Using LSTM as the classifier, this paper compares the classification results with and without the band combination 4–5. The results are shown in Table 7. The results in the table represent the average of the five test accuracies (Ave1) and the average after

removing the maximum and minimum values (Ave2), which eliminates the error caused by the experiment. The results show that the calculation results containing the 4–5 band combination are better (Table 8). Thus, the effect of the normalized difference index cannot be judged well by simply considering the variance, distance, or entropy between the overall tree species. Adding the ASDER calculation between tree species can provide more optimized features for tree species classification.

**Table 7.** Test results of LSTM.

| Exp | With Band_4–5 | Without Band_4–5 | Index by Var and Entropy |
|:---:|:---:|:---:|:---:|
| Ave1 | 88.58% | 87.53% | 87.78% |
| Ave2 | 88.45% | 87.08% | 87.34% |

**Table 8.** Accuracy results.

| Exp | MALSTM-FCN | LSTM | LSTM-FCN |
|:---:|:---:|:---:|:---:|
| Ave1 | 95.28% | 88.58% | 94.94% |
| Ave2 | 95.20% | 88.45% | 94.94% |

By sorting according to variance and entropy, this paper obtained 12 new band combinations, of which eight are consistent with the results of ASDER screening. Calculated with LSTM as the classifier, its Ave1 and Ave2 are 87.78% and 87.34%, respectively (Table 7), which are similar to the feature optimization results of ASDER. It indicates that good classification results can be obtained by simply relying on the feature optimization of ASDER.

In summary, ASDER is a feature optimization metric used for separate evaluation or joint evaluation of normalized difference features.

*5.2. MALSTM-FCN Structure Advantages*

To further verify the model's validity, the MALSTM-FCN model is compared with the LSTM-FCN and LSTM model results in this paper (Table 8 and Figure 11). It can be seen that MALSTM-FCN achieves better classification of individual tree species compared with LSTM-FCN and slightly higher overall accuracy, while using LSTM alone achieves less than 90% accuracy. The confusion matrix further demonstrates that W, Pi, and O are the difficulties in classification, which MALSTM-FCN improves.

Through previous experiments, this paper has demonstrated that the classification results can be optimized using a multi-head self-attention mechanism. This research investigates the effectiveness of the multi-head self-attention mechanism in improving model performance. Specifically, we compare the experimental results of different Multi-head Encoders while keeping the other model structures constant.

A Multi-head Encoder means that the newly generated feature under each time phase combines the features under all time phases. In this paper, selecting the feature under the last temporal phase as the output requires that the feature under the final temporal phase can represent the features of all temporal phases. For this kind of encoding, the multi-head self-attention mechanism is the core. In this paper, we compare the following cases: (1) encoding features and temporal sequences with the complete Transformer encoder block; (2) encoding of features and temporal sequences using only multi-head self-attention (Table 9). The difference between the first two is that the complete Transformer encoder block has more modules, such as residual linking, normalization, and multilayer perception, compared to the multi-head self-attention mechanism. To better reflect the difference between the two, this paper takes Ave2 as the accuracy to appropriately increase the number of training epochs and uses the model parameters with the highest test accuracy.

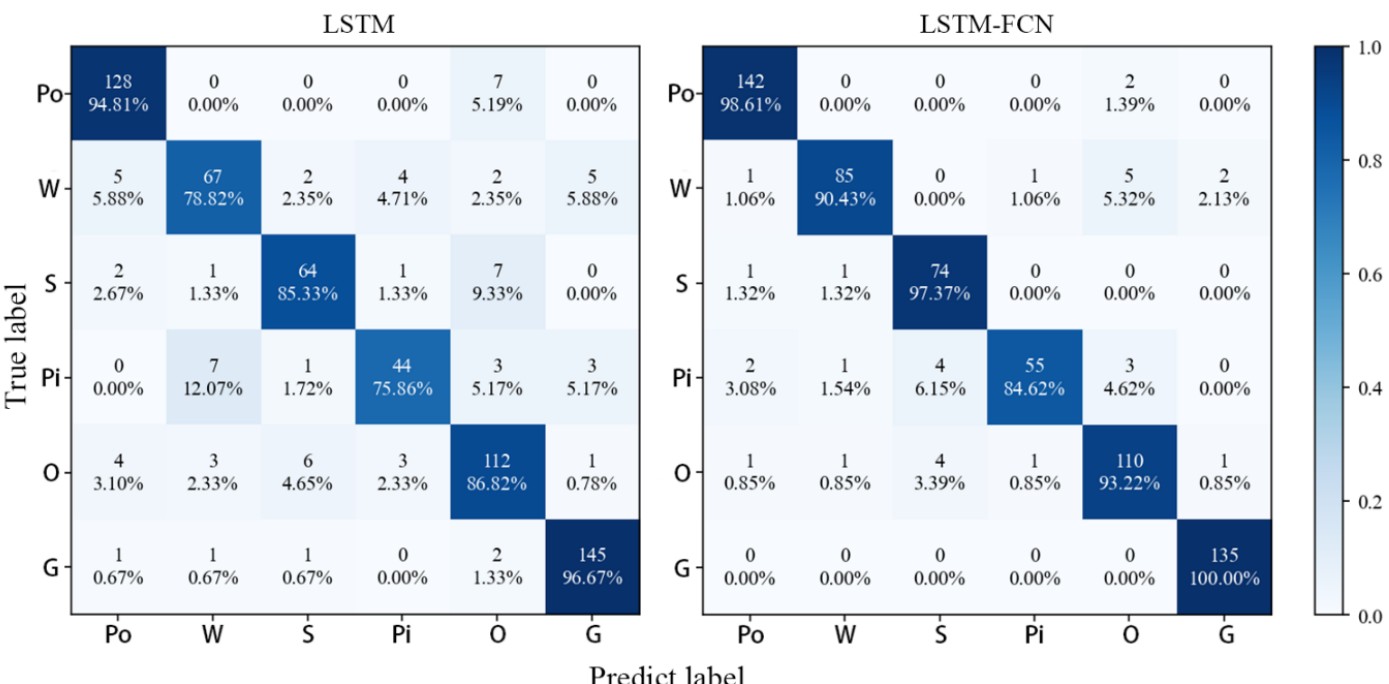

**Figure 11.** LSTM, LSTM-FCN confusion matrix.

**Table 9.** Branching experiments.

| Exp | Epoch | Acc |
| --- | --- | --- |
| only Multi-head Attention (Exp1) | 500 | 95.89% |
| total Transformer Encoder (Exp2) | 500 | 95.36% |

This paper conducted multiple tests and calculated the corresponding final accuracies. The experimental results with the median accuracy value were chosen to observe the changes in loss and accuracy (Figure 12). It was found that the other modules in the Transformer encoder do not exercise enough effect on the accuracy improvement. The simple use of a multi-head self-attention mechanism can exercise good effect. The loss values of both Exp1 and Exp2 converged to below 1.05, and Exp1 was slightly lower than Exp2. The Acc variation of Exp1 is basically concentrated between [0.955, 0.965]. The Acc variation of Exp2 was concentrated between [0.945, 0.955]. As a result, using the multi-head self-attention mechanism alone can have better results.

In addition, regarding the abrupt changes in the convergence process, this paper attributes to the dataset reasons. Due to the small sample data for training, large fluctuations tend to occur in the fitting process. This problem can be solved by extending the number of training epochs and reducing the learning rate. At the same time, this exposes the feature of insufficient samples.

This paper also compared classification experiments using LSTM branches alone. We found that the model's convergence is more difficult when using the MALSTM architecture alone. The model takes 900–1500 training epochs to converge to a reasonable level. This shows that only by combining FCN, can MALSTM converge faster and have better practicability.

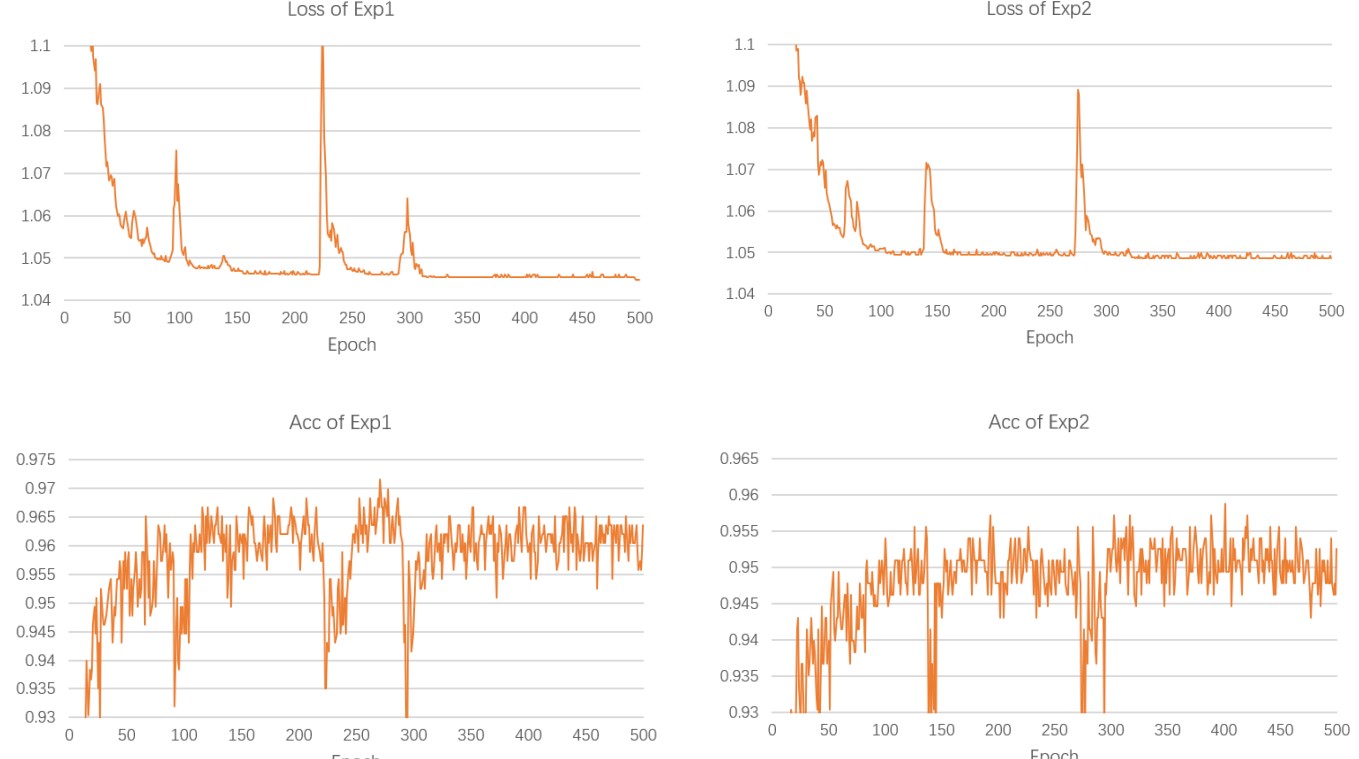

**Figure 12.** Loss and accuracy of the experiment.

### 6. Conclusions

This paper proposes the separability metric of ASDER and the time-series model of MALSTM-FCN for the classification refinement problem in the context of tree species classification. The main contributions of this paper are as follows:

1.  This paper proposed a separability metric ASDER based on standard deviation ellipse and angle weights. This metric provides a new separability evaluation standard for features constructed using the normalized difference method, which solves the problem of the lack of distinguishability in traditional separability metrics based on distance and variance in normalized difference features. This paper demonstrates the rationality of ASDER by deriving the dispersion degree between the sample points of different classes in a two-dimensional coordinate system. The experimental results demonstrate the effectiveness of incorporating the ASDER as a feature optimization criterion to improve classification accuracy.

2.  This paper presents the construction of the MALSTM-FCN model, with the addition of a multi-head self-attention mechanism to enhance the LSTM branch. The self-attention mechanism is utilized to compute the product of feature similarity between different temporal phases and the feature itself, thereby enhancing the correlation between features. The self-attention mechanism effectively addresses the problem of insufficient global consideration caused by the gradual loss of feature information from previous temporal phases during the computation process of LSTM. By using a multi-head perception method to obtain multi-layer semantic information between temporal features, this paper further improves classification accuracy.

In the discussion section, this paper compared the performance of the features evaluated based on ASDER and temporal variance for classification tasks. It was demonstrated that ASDER, as the sole feature optimization metric, can achieve good classification results. In terms of the model, this paper found that the multi-head self-attention mechanism is at the core of model enhancement, and adding a complete Transformer encoder does not improve classification performance. Moreover, this paper found that using only the MALSTM

branch for classification requires more computation rounds and has a limited improvement in accuracy. Therefore, combining MALSTM with FCN is necessary to achieve the best classification performance.

In addition, the sample size and research area size are essential to limit the further generalization of the above methods, and models to validate their transfer learning effects. This will be the focus of future research.

**Author Contributions:** Conceptualization, H.L. and D.M.; data curation, H.L. and D.M.; formal analysis, H.L., L.X. and X.L.; investigation, H.L.; methodology, H.L.; experiments and validation, H.L.; resources, D.M.; writing, H.L., D.M. and X.L. All authors have read and agreed to the published version of the manuscript.

**Funding:** This research has been jointly supported by the National Key R & D Program of China (2022YFB3903604) and the Fundamental Research Funds for the Central Universities (2-9-2021-045).

**Data Availability Statement:** Parts of the related data can be found at River Map (rivermap.cn, accessed on 4 March 2022), and Google Earth Engine (https://code.earthengine.google.com/, accessed on 22 March 2022).

**Acknowledgments:** We would like to thank Google for providing access to the Google Earth Engine platform, which allows us to access remote sensing data.

**Conflicts of Interest:** The authors declare no conflict of interest.

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
