# Peer review of "Tree Species Classification Based on ASDER and MALSTM-FCN"

_remotesensing, doi:10.3390/rs15071723_

Round 1

Reviewer 1 Report

The accurate classification of tree species plays an important role in forestry management. The author achieves accurate classification of forest tree species based on ASDER and MALSTMFCN, which is useful and interesting. However, only four tree species were classified in the paper, and the number of species used is relatively small. As stated in the paper, the park has more than 100 species totaling 530000 trees. If possible, it is suggested to use more tree species for tree species classification to verify the effectiveness of the method. Meanwhile, there are still some problems should be improved in the paper, as follows:

1. It’s showed that the non-time-series classification includes high-resolution images in Line 52-56, while in Line58 it’s stated that the time-series classification is mainly based on medium- and high-resolution satellites. Whether the high-resolution images belong to non-time series classification or time series classification? Suggest to explain it clearly.

2. In Line 118-119 “In summary, there is a trend to use object-based methods combined with phenological characteristics for tree species classification.” I don't understand how to draw this conclusion based on the previous content. In the previous content, there is less content about object-oriented classification and phenological characteristics. Suggest to explain it.

3. In Line 153 “the park has more than 100 species totaling 530,000 trees”. It’s showed that there are more than 100 tree species in the park. Why only 4 tree species were selected for classification instead of using more tree species? Does the method still maintain the high classification accuracy when more tree species are classified?

4. In Line 155-156 “Trees include oleander, juniper, acacia, dry willow, toon, and koelreuteria”. Whether the English translation of tree species is accurate. It is suggested to follow the corresponding Latin name after English name.

5. In Line 163 “ground truthing is carried out based on positioning techniques”. What positioning technology is used and how accurate the is positioning?

6. In Line 167 “a total of 2104 sample points”. What is the distance between sampling points? Does the distance meet the requirements of remote sensing image resolution?

7. The picture of Figure 3 is unclear. It is recommended to replace it with a clearer picture.

8. Suggest to change the Latin name of the tree species in Table 1 into English name.

9. In Line 197 “other typical bands is 10 m”. In Sentinel-2 image, except RGB band, only the spatial resolution of NIR band is10m. Suggest to replace other typical bands with NIR band.

10. Suggest to add relevant contents of accuracy evaluation to the Methods section.

11. In Line 206 “the time series features”. What features are included in the time series features? It is not explained in the paper. Suggest to add the relevant content.

12. In Line 210 “The first part is image preprocessing and segmentation”. Where is the specific content of this part? I don't see it in the paper. Suggest to add the relevant content.

13. In Line 314 “Where Band_2-Band_8, Band_11, and Band_12 represent…”. The spatial resolution of band11 and band12 are 20m, while the spatial resolution of band2 and band8 are 10m. How to deal with the difference in spatial resolution of different bands is suggested to be explained in the data preprocessing section.

14. In Line 327 “the rest of the indices”. What indexes are included in the rest of the indices? Suggest to add the relevant content in the appropriate section of the paper.

15. In Line 335 “In three of the classifications, the newly constructed indices gave better results.” What does three of the classifications mean? What are newly constructed indices? This can be seen from Table 3? Suggest to add the relevant explanations.

16. In Line 335 “the vegetation segmentation by GF2”. In addition to the data introduction, it seems that the GF-2 is only used to segment vegetation here. However, according to the technical flow diagram, GF-2 should not only be used to segment vegetation. Suggest to add the relevant content of GF-2 in the appropriate section of the paper.

17. Suggest to consider whether some content in discussion section should be moved to the experiment section.

Reviewer 2 Report

The paper is devoted to the tree species classification problem. This topic is relevant in the remote sensing data classification field, and the solution of the problem is in demand in the tasks of land management. The authors propose MALSTM-FCN network and ASDER rate for separating species on the Sentinel-2 and Gaofen-2 satellite images. New methods are used in the work and interesting results are obtained, but their description in the article needs to be improved.

1.      The Introduction is too broad in my opinion. You write about hyperspectral and LiDAR data, RF and SVM algorithms, which you do not use in your work. At the same time, almost no attention is paid to traditional separability metrics and, in particular, to the standard deviation ellipses. Please add this information in the Introduction section.

2.      The dataset has very little description. How many Sentinel-2 images were used, for what time periods, what bands were taken for calculations?

3.      Table 3 – if FP and FN rate is the number of misclassified species, you should add the whole number of samples of each type to this table or write the percentage of misclassified species.

4.      Figure 9 – the colors of the Populus and Willow classes are too similar, it is difficult to distinguish them in the image.

5.      Tables 6 and 7 – You write about 5 different tests, but do not provide information on how these tests differ.

6.      Line 391 – “after removing the most.” What most?

And finally, food for thought. You write that your dataset was randomly split into training and testing part. But random split can cause a situation where very spatially close pixels fall into different parts. Because of this, the accuracy can have high values,​ which sometimes will not correspond to the real land cover. It is better in your further work to collect test dataset separately, without spatial intersection with training part.

Round 2

Reviewer 1 Report

The authors addressed the main concerns, but extensive editing of English language is still required. It can be accepted after the extensive editing of English. 

Suggestion for the author: 

Before submitting materials, it's need to carefully check them. For example, in the Author Response File, there are two lines Chinese translation of English question about point 16. 

Author Response

Dear Reviewer,

Thank you very much for your advice and reminder. I am sorry for the problems that occurred in the previous submission. We have carefully analyzed the article's content and substantially rewritten its sentences.

We use the "revisions" feature to highlight revisions to a document for reviewers and editors.
Thanks again for your constructive comments on this article.

Reviewer 2 Report

Thank you for the detailed responses to my comments and excellent work with the paper. Good luck in further research!

Author Response

Dear Reviewer,
Thank you very much for your advice. We have carefully analyzed the article's content and substantially rewritten its sentences.
We use the "revisions" feature to highlight revisions to a document for reviewers and editors.
Thanks again for your constructive comments on this article. Good luck with your work.